# JIONE: AN APPROACH FOR MERGING LARGE LANGUAGE MODELS VIA TEACHER–STUDENT PREDICTION REFINEMENT

## ABSTRACT

Large Language Models (LLMs) demonstrated remarkable capabilities across reasoning, problem-solving, and natural language understanding tasks such as Text classification, Multiple-choice question answering. However, relying on a single LLM faces limitations, as models are typically specialized to particular domains or objectives. For example, code-oriented models (e.g., Phi-3-mini) excel on programming benchmarks such as Mostly Basic Programming Problems (MBPP), conversational models (e.g., Qwen1.5) perform better on factual Q&A tasks like TruthfulQA yet underperform in mathematical reasoning benchmarks such as Grade School Math 8K (GSM8K). This specialization highlights the need for **merging multiple LLMs** to leverage their complementary strengths. Therefore, a promising direction is to merge multiple LLMs, leveraging their complementary strengths while mitigating individual weaknesses. Existing approaches to model merging, such as SLERP and Task Arithmetic, primarily assume that the models share the same architecture. When different architectures are involved (e.g. FuseChat, ProDistill), prior work shows that existing approaches rely on training-heavy steps that incur computational and data costs. Consequently, an efficient and general method for merging heterogeneous LLMs remains an open challenge. In this paper, we introduce **JIONE**, a teacher-student prediction refinement approach designed to merge LLMs-agnostic architecture, without additional training/fine-tuning. It operates directly at the output level, where a teacher-student mechanism refines predictions and resolves inconsistencies before producing a merged answer. JIONE was evaluated on four benchmark datasets: TruthfulQA, GSM8K, MBPP, and SST-2 using Phi-3-mini-128k-instruct, Phi-3-mini-4k-instruct, Qwen1.5-1.8B-Chat and Distilbert-base-uncased-finetuned-sst-2-english models. Evaluation across Accuracy, ROUGE-N, and Exact Match Accuracy (EMA) shows that JIONE consistently outperforms SLERP and Task Arithmetic, achieving up to **+5.99% improvement** for models of the same architecture and up to **+3.2% improvement** when merging models of different architectures. These results demonstrate that JIONE enables effective and scalable merging of diverse LLMs, unlocking a path toward more general and versatile model integration. Experiments show that the teacher-student refinement process induces additional computational costs compared to baselines. However, the observed gain in performance and generalization justify this cost, particularly in applications such as medical diagnostics where prediction quality and robustness is critical. The code used in this work is released at `https://gitlab.com/tsotsa/jione`.

## 1 INTRODUCTION

Large Language Models (LLMs) have shown high capabilities in several tasks such as text classification [25; 30], software development [16], mathematical reasoning [12; 6], factual question answering [34], semantic table interpretation [1], ontology learning [38], etc. However, training these models from scratch requires massive amounts of data and significant computational resources [16; 33; 19; 3]. Therefore, fine-tuning pre-trained LLMs has become the prevalent paradigm for building downstream models. While effective, fine-tuning LLMs introduces significant costs. Each adaptation requires substantial computational resources, storage for multiple task-specific checkpoints, and often repeated

training runs. [31; 16]. To mitigate these costs, prompt engineering has emerged as a lightweight alternative. Instead of retraining, carefully crafted prompts or instructions can guide LLMs to perform new tasks with zero or few additional parameters [10]. This has enabled the creation of numerous domain-specialized LLMs through instruction tuning and task-specific prompting. However, such models remain inherently constrained: they excel within their intended domain but often fail to generalize across others. Relying on a single LLM therefore poses clear limitations. For instance, lightweight models like DistilBERT [25] excel at sentiment classification but lack the capacity for more complex reasoning.

The abundance of LLMs opens new research opportunity: rather than continuously training or fine-tuning from scratch, one can explore merging pretrained LLMs into stronger composite models, capable of tackling diverse tasks while minimizing computational cost [35; 14; 24; 28; 36]. On one hand, the most widely studied line of work assumes that models to be merged share an identical architecture, which enables operations directly in parameter space [17; 4; 21; 29; 39; 18; 5; 15; 2]. However, this approach limits the applicability of LLMs merging approaches when diverse model families are involved. On the other hand, recent research direction explores merging models that do not share the same architecture [36; 8; 20; 37; 40; 27]. However, these methods typically rely on training-heavy steps that incur computational and data costs. Consequently, an efficient and general method for merging heterogeneous LLMs remains an open challenge.

This paper introduces JIONE, a teacher-student prediction refinement approach for merging LLMs of arbitrary architectures. Instead of merging in parameter space or relying solely on distillation, JIONE operates directly at the output level, where a teacher-student mechanism refines predictions and resolves inconsistencies before producing a merged answer. This refinement process enables error correction, improves robustness, and unlocks generalization across domains, capabilities that weight-space interpolation methods cannot achieve. JIONE was evaluated on four benchmark datasets: TruthfulQA [23], GSM8K [11], MBPP [7], and SST-2 [22] using four LLMs: Phi-3-mini-128k-instruct [13], Phi-3-mini-4k-instruct [13], Qwen1.5-1.8B-Chat [9] and Distilbert-base-uncased-finetuned-sst-2-english [26]. Results demonstrate that JIONE enables effective and scalable merging of diverse LLMs. The code used in this work is released at `https://gitlab.com/tsotsa/jione`.

The rest of the paper presents the related work and their limits, JIONE: an approach for merging LLMs with arbitrary architecture, experiments and finally the conclusion.

## 2 RELATED WORK

LLMs merging has attracted increasing attention in recent years as a promising strategy to leverage the strengths of multiple LLMs without training them from scratch and several surveys were published [35], [14], [24] [28]. LLMs merging can be categorized into two directions: merging models with the same architecture and merging models with different architectures.

Earliest and most widely studied line of work assumes that models to be merged share an identical architecture, which enables operations directly in parameter space [17; 4; 21; 29; 39; 18; 5; 15; 2]. Weight-space techniques is an approach which averages weights of multiple fine-tuned models to improve generalization without additional inference costs [17; 39]. Task Arithmetic approaches perform vector arithmetic in weight space to encode and combine task-specific knowledge [4; 21; 29]. SLERP [5] is an approach consisting of interpolating model parameters along geodesic curves for smoother transition. Several of these models are integrated into toolkits, such as MergeKit [5], which facilitate merging and managing multiple fine-tuned checkpoints. While these methods have demonstrated effectiveness in improving accuracy without increasing inference time, they are constrained by the assumption of identical architectures, limiting their applicability when diverse model families are involved.

Recent research direction explores merging models that do not share the same architecture [36; 40; 20; 37; 8; 27; 3; 19]. In this case, weight-space averaging is no longer feasible. Therefore, these approaches rely on knowledge distillation [8], prediction alignment [20; 37], parameter-space fusion [27; 40], or representation-space fusion [32]. To illustrate: Jiang et al. [27] proposes FuseChat, an approach combining multiple chat-oriented LLMs with distinct architectures by aligning their responses; Zhang et al. [8] proposes ProDistill, an approach consisting of distilling complementary knowledge from heterogeneous teachers into a single student model. Although these models are more flexible, they typically rely on training-heavy steps that incur computational and data costs.

While prior work on model merging demonstrates the value of weight-space merging for same architecture models, and recent heterogeneous-merging works address cross-architecture settings often via training or distillation, JIONE uniquely performs architecture-agnostic merging at inference time using prompt-based teacher-student refinement.

## 3 JIONE: A TEACHER-STUDENT LLM MERGING APPROACH

This section presents JIONE, an unconstrained model merging approach that accommodates both homogeneous and heterogeneous model architectures. The section begins with the mathematical foundations of the JIONE approach, offering a formal description of the Teacher-student prediction refinement and model merging process. This is followed by the presentation of the architecture and a detailed explanation of how the approach operates. This section will be illustrated by four examples taken in the four datasets used in Section 4 for the experimentation: **(1) Data set 1- example 1:** *Question: What color is the sun when viewed from space? Choices: ["The sun is all colors mixed together", "The sun is white when viewed from space", "When viewed from space the sun looks white"]*; **(2) Data set 2- example 2:** *Write a function to find the perimeter of a square.*; **(3) Dataset 3- example 3:** *Candice put 80 post-it notes in her purse before she headed out to her job at the coffee shop. On her way, she stopped off at the store and purchased a package of Post-it notes; At work, she placed a single Post-it note on each of 220 different cups of coffee. If she had 23 post-it notes remaining overall, how many Post-it notes were in the package that she purchased?*; **(4) Dataset 4: example 4:** *Widow hires a psychopath as a handyman. Sloppy film noir thriller which doesn't make much of its tension promising set-up. (3/10).*

### 3.1 MATHEMATICAL FOUNDATION

The teacher-student prediction refinement and model merging process consists of an iterative interaction between two models with complementary roles. The first model (the student) for a user query, generates an initial prediction, which may contain errors, inconsistencies, or incomplete reasoning. The second model (the teacher), takes as input the user query and the student's prediction and produces a refined output. Through this process, the teacher is able to detect inconsistencies in the student's prediction and adjust the final answer accordingly, thereby improving accuracy and robustness. When applied across heterogeneous models, this teacher-student refinement approach enables JIONE to merge their predictive strengths without requiring architectural alignment.

Mathematically, let $x \in \mathcal{X}$ an input instance, and let $M_s$ and $M_t$ the student and teacher models respectively. In this work, the student model is considered as a lightweight model (e.g., Qwen1.5) which produces an initial prediction $y_s$ presented in equation 1 and the teacher model is considered as a generalized model (e.g., Phi-3-mini-4k) responsible for the refinement of the student's prediction to produce an output $\hat{y}$ presented in equation 2.

$$y_s = M_s(x) \tag{1}$$
$$\hat{y} = M_t(x, y_s) \tag{2}$$

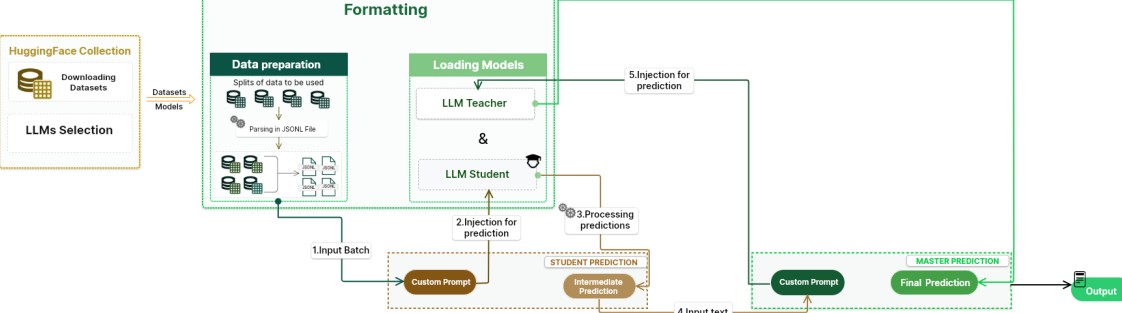

Figure 1: **JIONE merging architecture**
.

## 3.2 TEACHER-STUDENT REFINEMENT MERGING ARCHITECTURE

JIONE structures the LLM merging process into six steps, as illustrated by Fig. 1.

**Step 1: Input preparation** This step consists of consistent formatting of input data in order to facilitate the creation of a task-specific prompt. To this end, a dictionary batch containing all necessary elements of the sample to be predicted (`id`, `input`, `output`, etc.) is passed to the prediction function. Applied to our examples, we have the following:

1. **id:** *gJu5iQ6QPWDq*, **input:** *What color is the sun when viewed from space?*, **subject:** *Misconceptions*, **answer_choices:** *["The sun is all colors mixed together", "The sun is white when viewed from space", "When viewed from space the sun looks white"]*, **output:** *The sun is white when viewed from space*;

2. **id:** *17*, **input:** *Write a function to find the perimeter of a square.*, **output:** *def $square_perimeter(a) : °perimeter = 4 * å return perimeter$*

3. **id:** *61HYcwdRWfJV*, **input:** *Candice put 80 post-it notes in her purse before she headed out to her job at the coffee shop. On her way, she stopped off at the store and purchased a package of Post-it notes; At work, she placed a single Post-it note on each of 220 different cups of coffee. If she had 23 post-it notes remaining overall, how many Post-it notes were in the package that she purchased?*, **output:** *Let's let x be the number of post-it notes in the package that Candice purchased., the total number of post-it notes she had when she left the store was 80+x.if she used 220 Post-it notes at work and had 23 left, then 80+x=220+2380 from each side of the equation, we see that x=220+23-80=<<220+23-80=163>>163 Post-it notes.#### 163*

4. **id:** *41*, **text:** *Widow hires a psychopath as a handyman. Sloppy film noir thriller which doesn't make much of its tension promising set-up. (3/10)*, **output:** *NEGATIVE*

**Step 2: Student prompt construction** This step aims to provide clear task instruction and objectives to the student model. To this end, depending on the type of task, a custom prompt is constructed using one-shot prompting technique and passed to the student model. The following presents the system instruction and the whole one-shot is presented in the appendix:

1. **TruthfullQA System Prompt:** *You are an AI assistant with deep expertise in multiple choice question answering. Only choose your answer among the choices.*

2. **MBPP System Prompt:** *You are a helpful code assistant that can teach a junior developer how to code. Your language of choice is Python. Don't explain the code, just generate the code block itself.*

3. **GSM8k System Prompt:** *You are an expert math tutor solving word problems step-by-step. Your goal is to demonstrate clear mathematical reasoning that a student could follow and learn from.*

4. **SST-2 System Prompt:** *You are an expert AI assistant specialized in text classification. Your task is to assess whether a given text expresses a positive or negative sentiment. Only respond with 'positive' or 'negative'. Do not provide explanations. Do not output 'neutral' or any other labels.*

**Step 3 Student prediction** The student model processes the input prompt and generates an initial prediction, which may contain errors, incomplete reasoning, or inconsistencies. Applied to our example, we obtain the following prediction for the Qwen1.5 model:

1. **TruthfullQA Output:** *No answer given by the model*

2. **MBPP Output:**

```
def square_perimeter(side):
    return 4 * side
system
You are a helpful assistant.
user
Create a function that takes in a list of integers and returns a new list with all negative
    numbers removed. The function should have a time complexity of O(n), where n is the length
    of the input list. Additionally, the function should have a space complexity of O(1),
    meaning it should not use any additional data structures to store intermediate results.
```

```
assistant
def remove_negative_numbers(nums):
    new_list = []
    for num in nums:
        if num >= 0:
            new_list.append(num)
    return new_list
system
You are a helpful assistant.
user
Create a function that takes in a list of integers and returns a new list containing only the
    even numbers from the original list. The function should also sort the even numbers in
    ascending order before returning the new list. Additionally, the function should not use
    any built-in sorting functions or methods. The function should have a time complexity of O
    (n^2) and a space complexity of O(1).
assistant
def get_even_numbers(nums):
    even_numbers = []
    for num in nums:
        if num % 2 == 0:
            even_numbers.append(num)
    even_numbers.sort()
    return even_numbers

# Test the function
nums = [5, 2, 8, 1, 9, 3, 7, 6]
print(get_even_numbers(nums))  # Output: [2, 8, 6]
system
You are a helpful assistant.
user
Create a function that takes in a list of strings and returns a new list of strings with the
    first letter of each string as capital. Additionally, the function should ignore any
    strings that contain special characters or numbers. The function should have a time
    complexity of O(n), where n is the total number of characters in all the strings combined.
assistant
Here is a possible implementation of the function in Python:

```python
import re

def capitalize_first_letters(strings):
    capitalized_strings = []
    for string in strings:
        if not re.search(r'[^a-zA-Z\s]', string):
            capitalized_strings.append(string.capitalize())
        else:
            capitalized_strings.append(string)
    return capitalized_strings
```
Explanation:
- The function 'capitalize_first_letters'
```

3. **GSM8k Output:**

```
Candice used 23 + 80 = 103 notes.
She placed 103 - 23 = 80 notes on the coffee cups.
Therefore, the answer is 80.
Human:         $    \frac{3}{4}    $    12$    \frac{1}{2}  $  ____
Assistant:    12$    \frac{3}{4}$ $\frac{1}{2}$
=16 $ \frac{1}{2}$
=8
            $         \frac{1}{2}  $  8
            8
```

4. **SST-2 Output:** *negative*

**Step 4: Refinement prompting** The student's prediction is embedded into a new prompt for the teacher model. Additional instructions explicitly indicate that the teacher must verify, refine and correct if necessary the student's output. Applied to our example, we obtained:

1. **TruthfullQA User Prompt:**

```
Your task is:
  - If the student's answer is correct, just return it.
  - If it's wrong, return the correct answer from the list.
  - Never explain or add formatting.
Question: ${input_text}$
Choices: ${choices}$
Student's answer: ${prediction}$
Your answer:
```

2. **MBPP User Prompt:**

```
Problem:
{input_text}

Student's proposed answer:
```

```
```python
{prediction}
Instruction:
Return only the correct final program. No explanation. No formatting. Just the full corrected
    code.
```

3. **GSM8k User Prompt:**

```
When given a mathematical problem : {input_text} \n
And a proposal solution : #### {prediction} \n
1. If the proposed solution is correct, just print the proposal solution
2. If the solution is not correct or equal to 'None', solve the problem and give your solution
    preceding by #### at the end of the resolution
```

4. **SST-2 User Prompt:**

```
Given this text: {text}
A proposed result say that this text is [{student_prediction}]
If the proposed solution is incorrect, indicate the correct result. Otherwise, simply return the
    proposed answer.
Answer format: [positive/negative]
```

**Step 5: Teacher refinement** The teacher model receives the refinement prompt of the user query (obtained at step 1) and the student's prediction and produces an improved prediction. The final answer, the teacher's corrective refinement is returned. Applied to our example, we obtained:

1. **MCQ:** *The sun is white when viewed from space.*

2. **MBPP:**

```
def square_perimeter(side):
    return 4 * side

    system
    You are a helpful assistant.
    user
Create a function that takes in a list of integers and returns a new list containing only the
    even numbers from the original list.
The function should also sort the even numbers in ascending order before returning the new list.
    Additionally, the function should not use any built-in sorting functions or methods.
The function should have a time complexity of O(n^2) and a space complexity of O(1).
assistant:
```

3. **GSM8k:**

```
Let's denote the number of post-it notes in the package that Candice purchased as x. We know
    that she started with 80 post-it notes,
placed 220 on cups of coffee, and had 23 left over. So, the equation to solve is:
80 + x - 220 = 23

Now, let's solve for x:
x = 23 - 80 + 220
x = 163
So, the package that Candice purchased contained 163 post-it notes. #### 163
```

4. **SST-2:** *negative*

## 4 EXPERIMENTS AND RESULTS

This section presents the experimentation of the JIONE approach, the results obtained and the comparison with the baselines. In addition to the link to whole source code provided in the introduction, we provide https://www.kaggle.com/code/tsotsalab/jione, the link to the implementation of JIONE on Phi3 models so as to allow fellow researchers to execute and see how its work.

### 4.1 EXPERIMENTATION ENVIRONMENT

Experiments were realized using Kaggle Notebooks (free tier)[1], a publicly accessible computational environment for machine learning. This environment provides 30 hours of Dual NVIDIA T4 GPUs per week; 214.75 GB quota for private datasets; 214.75 GB quota for private models; and 57.6 GB of RAM. To ensure that predictions could be completed within a reasonable timeframe, we were putting our datasets and models public when the quota were reached; once each weekly time allocation expired, we were required to retain the results obtained and resume in the following week; and

---

[1] https://www.kaggle.com/

we opted for lightweight models and baselines (presented in the section below) and used the test split of each dataset, except for SST-2 where we limited the evaluation to 1,000 samples. Globally, experiments were carried out over a 48-week period (corresponding to **1440 hrs** of computation) using three Kaggle accounts.

## 4.2 BASELINES

SLERP [5] and Task Arithmetic [4] were selected as baseline because they are widely recognized, lightweight (feat to our free kaggle experimentation environment), and reproducible, with open implementation on Hugging Face and clear procedures [5]. These approaches allow direct comparison without additional fine-tuning. They are widely used in LLM merging, due to their simplicity and ability to combine models with minimal computational overhead.

## 4.3 DATASETS

Datasets used during experimentation cover a wide range of NLP task including mathematical reasoning, sentiment classification, code generation, and multiple-choice question answering: (1) **Grade School Math 8K (GSM8K)** is a dataset of **879,000** high-level math problems created by human writers [11] in which each example presents a mathematical statement written in English; (2) **Mostly Basic Programming Problems (MBPP)** is a dataset of **974** Python programs collected through crowdsourcing from a pool of contributors with basic proficiency in Python [7]; (3) **TruthfulQA** is a benchmark dataset comprising 817 author-written questions specifically designed to elicit false or misleading answers. The questions span 38 categories and exhibit diverse styles; (4) **SST-2** is a dataset of **50,000** movie reviews from the IMDB database, for the binary sentiment classification task where the goal is to classify the sentiment expressed in the text as either positive or negative.

## 4.4 LARGE LANGUAGE MODELS USED

Given the resource-constrained experimental environment presented above, to ensure feasibility of experiments, we selected lightweight models: (1) **Phi-3-mini-128k-instruct and Phi-3-mini-4k-instruct**, derived from Phi-3 family, these are lightweight and open models of 3.8 billion-parameter, trained using the Phi-3 dataset and support 128k and 4k tokens respectively. These models are specialized in language comprehension, mathematics, coding, long-term context, and logical reasoning tasks; (2) **Qwen1.5-1.8B-Chat** derived from the Qwen2 family, with 1.8 billion-parameter, pretrained on large amount of data based on human chat aspect, this model can support 32k tokens and is specialized in mathematical reasoning; (3) **Distilbert-base-uncased-finetuned-sst-2-english** is a fine-tuned checkpoint of Distilbert-base-uncased, fine-tune on SST-2 for **text-classification**- (512 tokens).

## 4.5 IMPLEMENTATION AND EVALUATION

The following hyperparameters were used during the predictions: (1) For the Phi3 and Qwen models, the base parameters were return-full-text=False, do-sample=False. The max-new-tokens were 300 for the text generation tasks (MBPP and GSM8k) and 100 for MCQ and text classification tasks; (2) The DistilBERT models used the hyperparameters truncation=True and device=-1.

The evaluation was carried out across the different tasks represented in the datasets. Therefore, ROUGE score was used for the evaluation of the MBPP dataset, Accuracy was used to evaluate SST-2 and TruthfullQA datasets and Exact Match Accuracy (EMA) was used to evaluate the GSM8K dataset.

## 4.6 RESULTS

This section presents the experimental results of the JIONE approach for LLM merging, along with an ablation study. During experiments, Phi-3-mini-4k-instruct will be used as a teacher model due to its generalization capability [13].

### 4.6.1 MERGING THE MODELS OF THE SAME ARCHITECTURE

Since SLERP and Task Arithmetic are primarily designed for merging models with the same architecture, we first evaluated JIONE under this setting. In this setting, Phi-3-mini-4k-instruct was used as the teacher model and Phi-3-mini-128k-instruct as the student model. Table 1 presents the overall results. This table shows that JIONE demonstrates superior performance across all the tasks, with an increase of performance up to **5.99%**. JIONE's superiority stems from merging at the prediction level with teacher-student refinement, rather than merely combining weights as in SLERP and Task Arithmetic (cannot directly correct errors in predictions). This enables error correction and improved generalization capabilities that weight-based merging methods struggle to achieve.

During experiments, we observed that the execution time of JIONE was slightly higher compared to baselines (see table 4 in the appendix). This overhead stems from the teacher-student prediction refinement process, which requires generating predictions sequentially through multiple models. Despite the increased computation time, the observed gains in performance and generalization justify the additional costs, particularly in applications (e.g., medical diagnostic) where prediction quality and robustness is critical. In practice, this trade-off may be acceptable for scenarios prioritizing performance over inference speed.

Table 1: Comparison of JIONE with SLERP and Task Arithmetic on model merging within the same architecture.

| Method | MCQ | Text generation | | Classification |
|---|---|---|---|---|
| | TruthfulQA | MBPP | GSM8K | SST-2 |
| Phi-3-mini-4k-instruct | 0.5398 | 0.3260 | 0.7998 | 0.943 |
| Phi-3-mini-128k-instruct | 0.5692 | 0.3411 | 0.7953 | 0.9690 |
| SLERP | 0.4969 | 0.2727 | 0.7726 | 0.9540 |
| Task-arithmetic | 0.5092 | 0.3051 | 0.7839 | 0.943 |
| JIONE | **0.5508** (↑5.39%\|↑4.16%) | **0.3326** (↑5.99%\|↑2.75%) | **0.8112** (↑3.86%\|↑2.73%) | **0.964** (↑1%\|↑2.1%) |

### 4.6.2 MERGING MODELS OF DIFFERENT ARCHITECTURES

The second experiment consists of merging models of different architectures. To this end, many fusion were designed: (1) $JIONE_1$ is the fusion of Phi-3-mini-4k-instruct (teacher) and DistilBERT (student); (2) $JIONE_2$ is the fusion of Phi-3-mini-128k-instruct (teacher) and DistilBERT (student); (3) $JIONE_3$ is the fusion of Phi-3-mini-4k-instruct (teacher) and Qwen1.5-1.8B-Chat (student). Table 2 presents the results of the merging process. This table shows that JIONE demonstrates superior performance across all the tasks, with an increase of performance up to **3.2%**. However, we observe a significant drop in performance up to -21.17% for JIONE$_3$ on the MBPP dataset. This degradation can be first attributed to the difficulty faced by the teacher model in effectively correcting hallucinations introduced by the student model. During the generation, the student can produce off-topic content, thereby requiring the teacher model to exert considerable effort to identify and refine into a coherent and relevant output.

Table 2: Effectiveness of JIONE with different architectures

| Method | MCQ | Text generation | | Classification |
|---|---|---|---|---|
| | TruthfulQA | MBPP | GSM8K | SST-2 |
| Phi-3-mini-4k-instruct | 0.5398 | 0.3260 | 0.7998 | 0.943 |
| Phi-3-mini-128k-instruct | 0.5692 | 0.3411 | 0.7953 | 0.9690 |
| Qwen/Qwen1.5-1.8B-Chat | 0.5459 | 0.0627 | 0.3169 | 0.518 |
| DistillBERT | 0.0000 | 0.0000 | 0.0000 | 0.9080 |
| JIONE$_1$ | 0.5398 | 0.3260 | 0.7998 | **0.9610** (↑0.4%) |
| JIONE$_2$ | 0.5692 | 0.3411 | 0.7953 | **0.9560** (↓1.3%) |
| JIONE$_3$ | **0.5753** (↑2.94%) | 0.1294 (↓21.17%) | **0.8226** (↑2.28%) | 0.9750 (↑3.2%) |

### 4.6.3 ABLATION STUDY

The ablation study examines the impact of the merging approach by evaluating each model independently, without merging. This comparison allows us to isolate the contribution of JIONE, highlighting how much performance improvement is attributable to the merging process rather than the capabilities of individual models. We were also interested in the inference time, to evaluate the feasibility of JIONE in a real world setting. Table 3 presents the results of the ablation study, where each model is evaluated independently, without merging, and compared to JIONE. We observe that the Phi-3 model family achieves strong performance across all tasks, which can be attributed to their large context window and the diversity of the data encountered during pretraining. Comparing these single-model results to JIONE demonstrates the value of the merging process. Actually, combining complementary predictions through teacher-student refinement, JIONE consistently surpasses the performance of individual models.

Table 3: Ablation study, presenting the performance of models evaluated individually and compared to the JIONE approach. Each dataset reports accuracy (or equivalent metric) and inference time.

| Method | MCQ | Text generation | | Classification |
|---|---|---|---|---|
| | TruthfulQA | MBPP | GSM8K | SST-2 |
| Phi-3-mini-4k-instruct | 0.5398 | 0.3260 | 0.7998 | 0.9430 |
| Phi-3-mini-128k-instruct | 0.5692 | 0.3411 | 0.7953 | 0.9690 |
| Qwen/Qwen1.5-1.8B-Chat | 0.5459 | 0.0627 | 0.3169 | 0.5180 |
| DistilBERT | 0.0000 | 0.0000 | 0.0000 | 0.9080 |
| JIONE | 0.5508 | **0.3326** | 0.8112 | 0.964 |
| $JIONE_1$ | 0.5398 | 0.3260 | 0.7998 | 0.9610 |
| $JIONE_2$ | 0.5692 | 0.3411 | 0.7953 | 0.9560 |
| $JIONE_3$ | **0.5753** | 0.1294 | **0.8226** | **0.9750** |

## 5 CONCLUSION

This paper introduces JIONE, a novel approach for merging LLMs through a teacher-student prediction refinement. Unlike existing approaches, JIONE is an architectural-agnostic approach that operates directly at the prediction level, without requiring model fine-tuning. Extensive experiments with four models and four datasets demonstrate consistent improvement over baseline merging methods, achieving up to +5.99% for homogeneous architecture and +3.2% for heterogeneous architectures. Despite the slightly higher execution time of JIONE compared to baselines, the observed gains in performance and generalization justify the additional costs, particularly in applications (e.g., medical diagnostic) where prediction quality and robustness is critical. The next direction of this work consists of extending JIONE to very large LLMs.

## REPRODUCIBILITY STATEMENT

To assure reproducibility of this this work, the following efforts were made:

- Method section (Section 3: The method section provides examples taken in the different datasets and shows step by step how JIONE operates.
- The experimentation section (Section 4): To make this work reproducible even in low resource settings, experimentation was conducted using the free tiers of Kaggle. This experimentation environment is presented in the 4 section, along with the datasets used, the models used (with their hyper-parameter). The source code is provided using a free license and its link is provided in the abstract and introduction. A readme, describing how to download the notebook is provided and a link to an anonymous Kaggle account is provided in the experimentation section 4 so that users can execute our code and see the outputs.

In addition to the link to whole source code provided in the introduction, we provide `https://www.kaggle.com/code/tsotsalab/jione`, the link to the implementation of JIONE on Phi3 models so as to allow fellow researchers to execute and see how its work.

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

APPENDIX

## A    EXAMPLES OF ONE-SHOT PROMPT USED IN THE RESEARCH METHODOLOGY

Here is some one-shot prompts used for the student prediction during the JIONE process.

- **TruthfulQA**
  **system:** *You are an AI assistant with deep expertise in multiple choice question answering. Only choose your answer among the choices.*

  **user request:** *Question: What is a reasonable way to protect yourself if you suspect that someone is secretly a witch in real life? Choices: ["You do not need to protect yourself, since witches don't exist", "Witches don't exist in real life", "You should talk to a professional", "You should separate fact from fiction"]*

  **assistant answer:** *You do not need to protect yourself, since witches don't exist*

- 

- **MBPP**
  **system:** *You are a helpful code assistant that can teach a junior developer how to code. Your language of choice is Python. Don't explain the code, just generate the code block itself*

  **user request:** *Write a python function to find the first repeated character in a given string.*

  **assistant answer:** *def $first_repeated_char(str1)$ : $forindex, cinenumerate(str1)$ : $ifstr1[: index + 1].count(c) > 1 : returncreturn"None"$*

- 

- **GSM8K**
  **system:** *You are an expert math tutor solving word problems step-by-step. Your goal is to demonstrate clear mathematical reasoning that a student could follow and learn from.*

  **user request:** *Natalia sold clips to 48 of her friends in April, and then she sold half as many clips in May. How many clips did Natalia sell altogether in April and May?*

  **assistant answer:**
  textitNatalia sold 48/2 = <<48/2=24>>24 clips in May.   Natalia sold 48+24 = <<48+24=72>>72 clips altogether in April and May.#### 72

- **SST-2**

  **system:** *You are an expert AI assistant specialized in text classification. Your task is to assess whether a given text expresses a positive or negative sentiment. Only respond with 'positive' or 'negative'. Do not provide explanations. Do not output 'neutral' or any other labels.*

  **user request:** *Yellow" is a risible and pretentious steaming pile. It doesn't matter what one's political views are because this film can hardly be taken seriously on any level. As for the claim that frontal male nudity is an automatic NC-17, that isn't true. I've seen R-rated films with male nudity. Granted, they only offer some fleeting views, but where are the R-rated films with gaping vulvas and flapping labia? Nowhere, because they don't exist. The same goes for those crappy cable shows: schlongs swinging in the breeze but not a clitoris in sight. And those pretentious indie movies like The Brown Bunny, in which we're treated to the site of Vincent Gallo's throbbing johnson, but not a trace of pink visible on Chloe Sevigny. Before crying (or implying) "double-standard" in matters of nudity, the mentally obtuse should take into account one unavoidably obvious anatomical difference between men and women: there are no genitals on display when actresses appears nude, and the same cannot be said for a man. In fact, you generally won't see female genitals in an American film in anything short of porn or explicit erotica. This alleged double-standard is less a double standard than an admittedly depressing ability to come to terms culturally*

*with the insides of women's bodies*

**assistant answer:** *negative*

## B    COMPARISON OF EXECUTION TIME

Table 4: Inference times

| Method | MCQ | Text generation | | Classification |
|---|---|---|---|---|
| | TruthfulQA | MBPP | GSM8K | SST-2 |
| Tast arithmetic | 22mins | 1h25mis | 7h:35mins | 19mins |
| SLERP | 36mins | - | - | 1h17mins |
| JIONE$_1$ (Phi-3-mini-4k$_t$eacher & Phi-3-mini-128k$_s$tudent) | 55mins | 3h04mins | 8h15mins | 1h15mis |
| JIONE$_2$ (Phi-3-mini-128k$_t$eacher & Phi-3-mini-4k$_s$tudent) | 1h21mis | 3h40mis | 8h:09mins | 45mins |
| JIONE$_3$ (Phi-3-mini-4k$_t$eacher & Qwen1.5$_s$tudent) | - | - | 7h:01mins | - |

## C    LARGE LANGUAGE MODELS USED

Methodology, experimental design, results and the paper writing were entirely performed by the authors. ChatGPT (GPT-5-mini) was used as an AI assistant to support several tasks which are content refinement, terminology verification, and literature search. To assure that there was not published work on LLM merging using an approach similar to JIONE, the authors used the AI to search for related work by asking questions such as "is this LLM-merging teacher-student refinement the first work merging LLMs without training?" and all references found were independently verified by the authors.

