# OpenReview forum: "JIONE: an Approach for Merging Large Language Models via Teacher–Student Prediction Refinement"
_ICLR.cc/2026/Conference — Submitted to ICLR 2026_

### Official Review · Reviewer_ciWt · 2025-10-27

**Soundness:** 3
**Presentation:** 3
**Contribution:** 3
**Rating:** 2
**Confidence:** 4

**Summary:**

This paper introduces JIONE, a novel teacher–student prediction refinement framework for merging multiple large language models (LLMs) without requiring training or architectural compatibility.
Unlike traditional parameter-space merging (e.g., SLERP, Task Arithmetic) or training-based distillation approaches (e.g., FuseChat, ProDistill), JIONE merges at the output level: a lightweight “student” model generates predictions, and a stronger “teacher” model refines these outputs through structured prompts.

The method is evaluated on four diverse tasks — TruthfulQA (factual Q&A), GSM8K (mathematical reasoning), MBPP (code generation), and SST-2 (sentiment classification) — using four LLMs (Phi-3-mini variants, Qwen1.5-1.8B-Chat, and DistilBERT).
Experiments show up to +5.99% accuracy improvement for same-architecture merging and up to +3.2% for heterogeneous architectures compared to SLERP and Task Arithmetic.

**Strengths:**

1. Introduces a training-free, prediction-level merging paradigm, distinct from weight-space and distillation methods.
2. Can merge models of different architectures (e.g., transformer vs. DistilBERT), expanding the scope of model fusion research.
3. Operates purely at the inference/prompt level; feasible even with limited compute.

**Weaknesses:**

1. The “mathematical foundation” section is superficial (only two equations) and lacks formal analysis of convergence, bias correction, or uncertainty propagation.
2. The paper equates sequential inference (student → teacher refinement) with “model merging,” but this is function composition, not true merging in the machine learning sense.
3. Once merged, JIONE does not produce a new model; every inference requires both models sequentially — not a genuine merge artifact.
4. Improvements could stem from the teacher model alone, not genuine cross-model fusion.
5. It’s unclear whether reversing teacher/student roles or swapping architectures would yield consistent results.
6. How are conflicts or contradictions between teacher and student handled? There is no formal reconciliation rule or confidence weighting.
7. Using free Kaggle T4 GPUs with small models weakens the claim of generality for “LLMs.” Results on small models may not extrapolate to real LLMs (>70B).
8. Improvements (e.g., +1–3%) are within noise margins; no confidence intervals, variance analysis, or multiple runs are reported.
9. For SST-2, only 1,000 samples used; for others, unclear how test data were sampled or randomized.
10. Ignores recent heterogeneous merging techniques like MergeMoE, Rebasin, or ModelSoups++ beyond 2024-2025 literature.
11. Execution time comparisons exist, but no quantification of FLOPs, memory footprint, or cost per query.
12. Baselines (SLERP, Task Arithmetic) are trivial and not state-of-the-art for heterogeneous merging.
13. Only four models; all relatively small; no multilingual, multimodal, or encoder-decoder tests.
14. ROUGE for MBPP (a code-generation task) is questionable; exact match or pass@k would be more appropriate.
15. Lacks example-based qualitative error analysis or human judgment verification.
...

**Questions:**

-

---

### Official Review · Reviewer_kVDQ · 2025-10-27

**Soundness:** 2
**Presentation:** 1
**Contribution:** 1
**Rating:** 2
**Confidence:** 4

**Summary:**

The paper attempts to explore an interesting idea, but the overall execution and clarity are quite weak. The motivation is not clearly stated, and the connection between the problem and proposed method feels forced. The writing is difficult to follow, with many vague descriptions and missing technical details. Experimental design lacks rigor — datasets, baselines, and evaluation metrics are poorly justified. Results are presented without sufficient analysis, and the claimed improvements are marginal or unconvincing.

**Strengths:**

The topic is somewhat relevant to current research trends.

The general direction could be valuable if properly developed.

**Weaknesses:**

- The paper is hard to read, with inconsistent terminology and unclear mathematical formulations.

- The "teacher–student" setup is a trivial two-step generation, offering no real innovation over self-consistency or ensemble methods.

- Baselines are inappropriate or missing; evaluation metrics are inconsistent and sometimes incorrect (e.g., using ROUGE for code generation).

- Reported improvements are small or even negative in some cases, with no solid analysis to explain them.

**Questions:**

Refer to weakness

**Details Of Ethics Concerns:**

The submitted code repository is not anonymized and reveals author identity, violating double-blind review policy.

---

### Official Review · Reviewer_4rPE · 2025-10-31

**Soundness:** 2
**Presentation:** 1
**Contribution:** 2
**Rating:** 2
**Confidence:** 3

**Summary:**

This paper introduces JIONE, a novel, training-free, and architecture-agnostic method for merging Large Language Models (LLMs). The core idea is to use a teacher-student framework at inference time. A "student" model first generates an initial prediction for a given input. This prediction, along with the original input, is then passed to a "teacher" model, which is prompted to verify, correct, and refine the student's output to produce the final answer.

**Strengths:**

*   **Originality and Simplicity:** While teacher-student frameworks and output refinement are not new concepts, their application as a direct, training-free *model merging* technique is a novel and clever framing. Instead of complex operations in parameter or representation space, JIONE leverages the instruction-following capabilities of modern LLMs to perform merging at the semantic, output level. This simplicity is a significant asset.

*   **Significance and Practicality:** It addresses a critical challenge in the field: how to effectively combine the complementary strengths of diverse, often architecturally-heterogeneous LLMs without incurring the massive costs of retraining or complex distillation setups. JIONE offers a plug-and-play solution that is immediately applicable, lowering the barrier to creating more powerful and robust composite AI systems. Its effectiveness in a low-resource setting (Kaggle free-tier) further underscores its accessibility.

**Weaknesses:**

*   **"Refinement" vs. "Fallback" Mechanism:** The term "prediction refinement" may not fully capture the mechanism in all cases. The provided examples (e.g., for TruthfulQA and GSM8K) suggest that when the student model produces a nonsensical or empty answer, the teacher model effectively ignores it and solves the problem from scratch. This is more of a "fallback" or "re-solve" mechanism rather than a true refinement of an existing, partially-correct answer. The paper would be stronger if it acknowledged this distinction and perhaps analyzed the frequency of true refinement versus fallback.

*   **Insufficient Baselines for Heterogeneous Merging:** This is the most significant weakness of the evaluation. While SLERP and Task Arithmetic are appropriate baselines for same-architecture merging, the paper lacks a comparative baseline for the heterogeneous case. The JIONE results are only compared against the performance of the individual models. A simple, training-free ensemble baseline is missing. For example, what would be the performance of a "judge-based" ensemble, where both models generate an output and the teacher model is simply asked to pick the better one? Or a simple voting mechanism for classification? Without such a comparison, it is difficult to ascertain whether the proposed teacher-student *refinement* structure is uniquely effective, or if the gains come from any simple two-model interaction.

*   **Misleading "Ablation Study":** The study presented in Table 3 is labeled as an "Ablation Study," but it does not ablate any component of the JIONE method itself. It is a restatement of the performance of JIONE versus its constituent single models. A true ablation study would investigate the impact of specific components of the JIONE framework. For example, it could test a variant where the teacher prompt does not include the student's prediction, to isolate the effect of seeing a prior (potentially flawed) attempt.

*   **Limited Scale and Generalizability:** The experiments are conducted on relatively small models (under 4B parameters). While this is understandable given the stated resource constraints, the broad claims about "unlocking a path toward more general and versatile model integration" should be tempered. It remains an open question whether the same dynamics and performance gains would hold with much larger, state-of-the-art models, where the "student" might be far more capable and the nature of errors more subtle.

**Questions:**

1.  **Clarification on Results in Table 1:** In Table 1, what do the two percentage improvements in parentheses, e.g., `(↑ 5.39% | ↑ 4.16%)`, specifically refer to? Is it the improvement over the two baselines (SLERP and Task Arithmetic) respectively, or over the two individual models? Please clarify this notation.

2.  **On the Nature of "Refinement":** Could you comment on the distinction between true "refinement" (improving a partially correct answer) and "fallback" (re-solving after a poor student answer)? Did you analyze how often the teacher model appeared to build upon the student's output versus simply disregarding it? This could provide deeper insight into JIONE's inner workings.

3.  **Regarding Baselines for Heterogeneous Merging:** Could you justify the absence of a simple ensemble baseline for the heterogeneous merging experiments? A comparison against a "best-of-N" or "judge-based" ensemble, where the teacher model selects the best response from the student and its own potential response, would significantly strengthen the claim that the refinement pipeline itself is the key contributor to the performance gain.

4.  **Regarding the Ablation Study:** Have you considered a true ablation where the teacher model is prompted to solve the task without being shown the student's prediction? This would help disentangle the benefits of the two-step, structured prompting from the specific benefit of having the teacher correct a student's answer.

5.  **Failure Mode on MBPP:** The performance drop for JIONE 3 on the MBPP task is a fascinating and important result. You attribute it to the teacher's difficulty in correcting the student's hallucinations. Does this suggest a fundamental risk where a sufficiently low-quality student can act as a "distractor," actively harming the performance of a more capable teacher? Have you considered adding instructions to the teacher prompt to explicitly "ignore the student's answer if it is nonsensical or completely off-topic" to mitigate this?

---

### Meta-Review · Area_Chair_TvGb · 2025-12-30

**Summary:**

The paper proposes a teacher-student inference pipeline framed as a model merging technique, but the reviewers unanimously found the methodology trivial and the experimental validation insufficient. The authors did not submit a rebuttal to address these issues. Hence, I recommend rejecting this submission.

**Reviewer Concerns:**

The authors did not provide a rebuttal and engage in the discussion phase. Hence, the raised concerns by reviewers are still unsolved. Reviewer 4rPE raised significant concerns about the lack of appropriate baselines for heterogeneous merging, e.g., simple ensembles, and questioned whether the method performs true refinement or simply acts as a fallback mechanism. Reviewer KVDQ criticized the execution as weak and the method as a trivial two-step generation that offers no real innovation over existing self-consistency or ensemble methods. Reviewer ciWt argued that sequential inference is merely function composition rather than true model merging and noted a lack of statistical rigor, such as confidence intervals. Reviewer bZQe (and others) flagged that the provided code repository contained author names, directly violating the double-blind review policy.

**Reviewer Scores:**

Based on the above issues, I think all the reviewers would keep their scores.

---

### Decision · Program_Chairs · 2026-01-26

Reject